**Data Availability Statement:** All relevant data are within the manuscript and its Supporting Information files (SPSS )

**Funding:** The author(s) received no specific funding for this work.

# Pre-pregnancy maternal BMI as predictor of neonatal birth weight

**Rafia Gul**[1]*, **Samar Iqbal**[1], **Zahid Anwar**[1◐], **Saher Gul Ahdi**[2◐], **Syed Hamza Ali**[1‡], **Saima Pirzada**[1‡]

**1** Department of Pediatrics, Fatima Memorial Hospital, Lahore, Shadman, Pakistan, **2** Department of Pediatrics, Combined Military Hospital, Lahore, Pakistan

◐ These authors contributed equally to this work.
‡ These authors also contributed equally to this work.
* docrafiagul@gmail.com

## Abstract

### Introduction

BMI is a tool to measure maternal nutritional status. Maternal malnutrition is frequently reported health problem especially during child bearing age and effects neonatal birth weight.

### Aim

To determine relationship between prepregnancy maternal BMI and neonatal birth weight.

### Methods and material

Prospective, cross sectional study conducted in Fatima Memorial Hospital, Lahore, Pakistan over a period of 1 year including 2766 mother—neonate pairs. All full term, live born neonates of both gender in early neonatal period (<72 hours) with documented maternal pre-pregnancy and/or first trimester BMI were enrolled. Data analysis using SPSS version 20, was performed.

### Results

Data analysis of 2766 mother–neonates pairs showed that there were 32.9% overweight and 16.5% obese mothers. More than two third of all overweight and obese mothers were of age group between 26–35 years. Diabetes mellitus, hypertension, medical illness, uterine malformations and caesarean mode of delivery were more prevalent in obese mothers as 22.8%, 10.1%, 13.2%, 2.6% and 75.4% respectively. Mean birth weight, length and OFC increased with increasing maternal BMI. Comparing for normal weight mothers, under-weight mothers were at increased risk of low birth weight (p< 0.01) and low risk of macrosomic neonates (p<0.01). However overweight and obese mothers were comparable to normal weight mothers for delivering macrosomic neonates (p 0.89 and p 0.66 respectively).

**Competing interests:** The authors have declared that no competing interests exist.

## Conclusions

Our study highlights that direct relationship exists between maternal BMI and neonatal birth weight.

## Introduction

Maternal health status is a key determinant of foetal growth. Maternal malnutrition refers to deficiency, excess or imbalance in maternal intake of energy and/or nutrients [1]. Body mass index (BMI) is a globally accepted gauge to assess maternal nutritional status. It is calculated as weight in kilogram divided by the square of height in meter ($kg/m^2$).

According to recommendations by Institute of Medicine (IOM) adapted by Pediatrics and Pregnancy Nutrition Surveillance System (PNSS), maternal BMI is classified as underweight (BMI <18.5), normal (BMI = 18.5 to 24.9), overweight (BMI $\geq$ 25.0 to 29.9) and obese (BMI $\geq$ 30) [2].

During childbearing age malnutrition, both under-nutrition, and obesity, is a global health problem. The developed countries share an increasing burden of overweight and obesity. Like in the USA, 4% of all American women are underweight, 47% normal, and 48% overweight [3]. Under-nutrition was once considered as a red flag in developing countries like in Pakistan. However, there is a change in trend and now Pakistan ranks among the top 10 countries where obesity is prevalent. About 23.9% and 6.3% of Pakistani females of childbearing age are overweight and obese respectively [4].

Maternal and fetal wellbeing is directly coupled. Optimal fetal growth is influenced by a number of factors. These factors through an intricate mechanism control fetal metabolic signaling pathways and guide "fetal programming". Maternal nutritional status is most important among these factors as it ensures continuous nutrients supply to developing fetuses. Other factors are diabetes mellitus, hypertension, anemia, smoking, chronic illness, uterine problems, periodontal health, drugs, addiction, weight gain during pregnancy, parity, and the number of fetuses [5–7].

Growth assessment of neonate at birth is done by measuring weight, length, and occipito-frontal circumference (OFC) [8].

To author's best knowledge, limited data of the Southeast Asian population in the context of IOM and PNSS guidelines for maternal BMI classification and neonatal birth weight is available. In particular, there is a paucity of studies conducted on Pakistani women. In a small study conducted in Islamabad, among 164 of total deliveries, 10% of low birth weight babies were born to mothers with BMI>25 [9]

The nutritional status of females in childbearing age, predicted by BMI, is a significant indicator of neonatal birth weight. Our study aimed to identify the relationship of maternal BMI with neonatal birth weight.

## Material and methods

Study was conducted after ethical approval from IRB committee, Fatima Memorial Hospital, Lahore. Written Consent was taken from individual mothers for each mother-neonate pair in study.

This was a prospective, cross sectional study conducted in the Department of Neonatology, Fatima Memorial Hospital, Shadman Lahore, from November 2018 till October 2019. Data was screened for maternal-neonatal pairs.

Full term (born between $37^{+0}$ to $41^{+6}$ weeks complete gestation) live born neonates of both gender were enrolled in this study. All neonates were assessed for their growth parameters within 72 hours. Birth weight was measured using digital weight scale and categorized as low birth weight ($< 2.5$ kg), normal birth weight (2.5 to 4.0kg) and macrosmic ($> 4.0$kg) babies. Neonatal length and OFC were measured using infant-meter and OFC measuring tape respectively.

Maternal demographic data was collected from antenatal record documented on special hospital health card for each pregnant female. Only those mothers with documented maternal pre-pregnancy and/or first trimester BMI record were enrolled in this study. First trimester maternal weight was also acceptable as there is usually minimal weight gain during this period. We calculated maternal BMI as the main exposure and classified using Paediatrics and Pregnancy Nutrition Surveillance System (PNSS) as underweight (BMI $<18.5$), normal weight (BMI = 18.5 to 24.9), overweight (BMI $\geq 25.0$ to 29.9) and obese (BMI $\geq 30$).

The covariates maternal age, parity, gestational age, residence, mode of delivery (SVD–spontaneous vaginal delivery, instrumental or LSCS–low segment caesarean section), diabetes mellitus, hypertension, chronic medical illness, gestational weight gain, uterine malformation, poor periodontal health, anemia and blood transfusion, number of foetuses, drugs, addiction and smoking were addressed while calculating relationship between prepregnancy maternal BMI and neonatal birth weight.

WHO has defined optimal weight gain during pregnancy for each BMI group. For under weight it is 12 – 18kg, normal weight 11.5 – 16kg, over weight 7–11.5 kg and obese 5–9 kg [2]

All maternal–neonatal pairs were excluded with incomplete maternal or neonatal data, premature, neonatal age $> 72$ hours of life at first examination or refusal to participate.

## Statistical analysis

All relevant data were recorded using a proforma in hand-writing and statistically analysed electronically using SPSS v20.

Maternal covariates for each BMI group were presented as descriptive statistics (numbers and percentages). Neonatal birth weight, length and OFC were presented as the means ± standard deviation for each maternal BMI group.

Multiple logistic regression model was used to correlate prepregnancy maternal BMI (primary exposure) and neonatal birth weight (primary outcome) while controlling all possible covariates / risk factors. Pairwise calculations were made for all BMI categories in form of mean difference and 95% confidence intervals (CI) while normal maternal BMI (18–24.9) was taken as reference category.

We performed analysis of variance (ANOVA) to test difference in mean birth weight for each maternal BMI group. Statistically significant difference among all BMI groups for mean birth weight was done by using Games-Howell post-hoc test and p-value less than 0.05 was taken as significant.

According to each maternal BMI group, descriptive statistics (number and frequency) was done for neonatal birth weight groups as low birth weight ($<2.5$kg), normal birth weight (2.5 – 4kg) and macrosmic ($>4$kg) babies.

## Results

In present study 3390 mother—neonate pairs were enrolled initially but 624 were excluded (300 mothers were unaware of their pre-conceptional weight, 204 refused to participate in study, 75 mothers have incomplete data and 45 mother–neonatal pair presented after 72 hours of life).

Nearly 46.6% of the study population (n = 1290) included births to mothers with normal BMI. This was taken as the reference group to which other BMI groups were compared while using multivariate logistic regression. In the study population, 4.0% (n = 110) of the babies were born to mothers with underweight, 32.9% (n = 910) to overweight and 16.5% (n = 456) to obese mothers. Table 1 highlights the percentages and numbers of all mothers according to BMI categories for all covariates (maternal characteristics).

More than two third of all overweight and obese mothers were of age group between 26–35 years. Majority (93.4%) of study was consisted of urban population. Diabetes mellitus, hypertension, medical illness and uterine malformation were commonly documented in obese mothers as 22.8%, 10.1%, 13.2% and 2.6% respectively (Table 1). Periodontal health problems, anemia and blood transfusion during pregnancy were observed in both extremes of BMI (Table 1). Only 30.7% obese mothers completed full term gestation (39–40 weeks) as compared to 39.7% normal weight mothers (Table 1). The trend towards caesarean delivery with increasing BMI and noted to be 77.5% in overweight and 75.4% in obese mothers.

The results of multivariate linear regression analysis models, with birth weights as the dependent variables and the maternal BMI categories as the exposure variables controlling for study covariates, have been tabulated as Table 2.

Our best esteem of difference show that both normal weight and underweight mothers are statistically equal for having low birth weight babies (p < 0.96, 95%CI -0. 046 to 0.067). However in overweight and obese mothers, incidence of low birth babies was low as compared to normal weight mothers (p < 0.01, 95%CI -0.106 to -0.055 and p < 0.01, 95%CI -0.122 to -0.060) respectively.

After adjusting for potential confounders, underweight mothers were less likely to have macrosomic neonates as compared to mothers with BMI 18.5–24.9 (p < 0.01, 95%CI -0.091 to 0.000). However there was no statistically significant difference between normal weight, overweight and obese mothers for prevalence of macrosomic neonates (p 0.89, 95%CI -0.026 to 0.014 and p 0.66, 95%CI -0.021 to 0.029) respectively.

Among all delivered neonates, there were 1468 male and 1298 female neonates. One-way analysis of variance (ANOVA) was conducted to evaluate the null hypothesis that maternal BMI has no effect on neonatal birth weight (n = 2766). The independent variable maternal BMI has 4 groups as underweight (M = 2.98, SD = 0.66, n = 110), normal weight (M = 3.13, SD = 0.62, n = 1290), overweight (M = 3.21, SD = 0.573, n = 910) and obese (M = 3.24, SD = 0.57, n = 456).

The assumption of normality was tenable to all 4 BMI groups. The assumption of homogeneity of variance was tested using Levene test and found not be tenable $F_{(3, 2762)} = 4.96$, p = 0.002. However, robust tests of equality of means were found to be tenable by both Welch {$F_{(3, 462.50)} = 7.76$, p = 0.000} and Brown- Forsythe {$F_{(3, 634.39)} = 7.86$, p = 0.000}.

Post hoc comparison to evaluate pairwise difference among group means were conducted with the use of Games-Howell test since equal variance was not tenable. Test revealed significant pairwise difference in mean neonatal birth weight of undernourished and normal weight and overweight and obese mothers (p< 0.05).

However mothers with BMI <18.5 do not differ significantly from BMI = 18.5 to 24.9 (p>0.05). Similarly statistically insignificant difference was noted in overweight (BMI ≥ 25.0 to 29.9) and obese (BMI ≥ 30) (p>0.05).

Mean values of neonatal weight, OFC and length have been tabulated according to maternal BMI group in Table 3.

According to each maternal BMI group, descriptive statistics (number and frequency) was done for neonatal birth weight groups as low birth weight (<2.5kg), normal birth weight (2.5 – 4kg) and macrosmic (>4kg) babies in Table 4.

**Table 1. Data of maternal characteristics according to BMI groups.**

| Maternal characteristics | Pre pregnancy maternal BMI kg/m² n = 2766 | | | | |
|---|---|---|---|---|---|
| | **Underweight (BMI <18.5)** | **Normal weight (BMI 18.5 to 24.9)** | **Overweight (BMI ≥ 25.0 to 29.9)** | **Obese (BMI ≥ 30)** | **Total** |
| | **110 (4.0%)** | **1290 (47%)** | **910 (33%)** | **456 (16%)** | |
| **Maternal age (years)** | | | | | |
| ≤ 25 | 44 (40.0%) | 414 (32.1%) | 186 (20.4%) | 98 (21.5%) | 742 (26.8%) |
| 26–35 | 66 (60.0%) | 806 (62.5%) | 634 (69.7%) | 330 (72.4%) | 1836 (66.4%) |
| 36–40 | 0 (0.0%) | 70 (5.4%) | 86 (9.5%) | 28 (15.2%) | 184 (6.7%) |
| >40 | 0 (0.0%) | 0 (0.0%) | 4 (0.4%) | 0 (0.0%) | 0 (0.1%) |
| **Parity** | | | | | |
| Primiparous | 32 (29.1%) | 584 (45.3%) | 318 (34.9%) | 144 (31.6%) | 1078 (39.0%) |
| 2–5 | 78 (70.9%) | 702 (54.4%) | 592 (65.1%) | 312 (68.4%) | 1684 (60.9%) |
| >5 | 0 (0.0%) | 4 (0.3%) | 0 (0.0%) | 0 (0.0%) | 4 (0.1%) |
| **Gestational age (weeks)** | | | | | |
| 37–38 | 70 (63.6%) | 758 (58.8%) | 576 (63.3%) | 314 (68.9%) | 1718 (62.1%) |
| 39–40 | 40 (36.4%) | 510 (39.5%) | 320 (35.2%) | 140 (30.7%) | 1010 (36.5%) |
| 41–42 | 0 (0.0%) | 22 (1.7%) | 14 (1.5%) | 2 (0.4%) | 38 (1.4%) |
| **Residence** | | | | | |
| Urban | 110 (100.0%) | 1210 (93.8%) | 842 (92.5%) | 422 (92.5%) | 2584 (93.4%) |
| Rural | 0 (0.0%) | 80 (6.2%) | 68 (7.5%) | 34 (7.5%) | 12 (6.6%) |
| **Mode of delivery** | | | | | |
| SVD | 30 (27.3%) | 322 (25.0%) | 100 (11.0%) | 70 (15.4%) | 482 (17.4%) |
| Instrumental | 6 (5.5%) | 51 (4.0%) | 105 (11.5%) | 42 (9.2%) | 176 (6.4%) |
| LSCS | 74 (67.3%) | 917 (71.1%) | 705 (77.5%) | 344 (75.4%) | 2108 (76.2%) |
| **Smoking** | 2 (1.8%) | 6 (0.5%) | 4 (0.4%) | 0 (0%) | 12 (0.4%) |
| **Diabetes Mellitus** | 8 (7.3%) | 122 (9.5%) | 108 (11.9%) | 104 (22.8%) | 342 (12.4%) |
| **Hypertension** | 0 (0)% | 88 (6.8%) | 202 (22.2%) | 46 (10.1%) | 336 (12.1%) |
| **Medical illness** | 0 (0%) | 114 (8.8%) | 88 (9.7%) | 60 (13.2%) | 262 (9.5%) |
| **Adequate weight gain during pregnancy** | 21 (19.1%) | 184 (14.3%) | 147 (16.2%) | 89 (19.5%) | 441 (15.9%) |
| **Uterine malformation** | 0 (0%) | 42 (3.3%) | 24 (2.6%) | 12 (2.6%) | 78 (2.8%) |
| **Periodontal health problems** | 44 (40.8%) | 408 (31.6%) | 268 (29.5%) | 156 (34.2%) | 876 (31.7%) |
| **Drugs & addiction** | 0 (0%) | 8 (0.6%) | 24 (2.6%) | 10 (2.2%) | 42 (1.5%) |
| **Anemia & blood transfusion during pregnancy** | 44 (40.8%) | 418 (32.4%) | 240 (26.4%) | 150 (32.9%) | 852 (30.8%) |
| **Singleton pregnancy** | 94 (85.5%) | 1252 (97.1%) | 910 (100%) | 440 (96.5%) | 2696 (97.5%) |

## Discussion

Our retrospective cross-sectional study helped in determining the impact of maternal BMI on neonatal birth weight in a tertiary care hospital.

Dudenhausen JW et al found in their study that before pregnancy, 4% of all American women were underweight, 47% were normal, and 48% were overweight [3]. According to Calik et al, prepregnancy overweight and obesity was documented in 20.6% and 3.9% pregnant

**Table 2. Pairwise comparisons of BMI groups for neonatal birth weights.**

| Neonatal birth weight | (I) Prepregnancy maternal BMI (18.5–24.9 (kg/m²) | (J) Prepregnancy maternal BMI (kg/m²) | p value | Mean Difference (I-J) | 95% Confidence Interval for Difference [b] | |
|---|---|---|---|---|---|---|
| | | | | | Lower Bound | Upper Bound |
| <2.5 | ref | <18.5 | 0.96 | .010 | -.046 | .067 |
| | | 25–29.9 | <0.01 | -.081* | -.106 | -.055 |
| | | ≥ 30 | <0.01 | -.091* | -.122 | -.060 |
| >4 | ref | <18.5 | <0.01 | -.045* | -.091 | .000 |
| | | 25–29.9 | 0.89 | -.006 | -.026 | .014 |
| | | ≥ 30 | 0.66 | .004 | -.021 | .029 |
| 2.5–4 | ref | <18.5 | 0.63 | .035 | -.035 | .105 |
| | | 25–29.9 | <0.01 | .087* | .056 | .118 |
| | | ≥ 30 | <0.01 | .088* | .049 | .126 |

• Prepregnancy maternal BMI (18.5–24.9) is reference category.

• P-values obtained from Poisson regression models.

• Model was based on estimated marginal means significant (*) at p = 0.05.

• [b] Model adjusted for covariates maternal age, parity, gestational age, residence, mode of delivery, diabetes mellitus, hypertension, chronic medical illness, gestational weight gain, uterine malformation, poor periodontal health, anemia and blood transfusion, number of foetuses, drugs, addiction and smoking.

females respectively [10]. However studies conducted in Pakistan and Bangladesh have shown that there is a trend towards over weight (23.9% and 40.1%) and obesity (6.3% and 21.2%) respectively [4, 11]. Our study shows similar trend towards over weight and obesity. For every underweight mother, there were 8 overweight and 4 obese mothers. This increasing prevalence of BMI can be due to fact that all study population belonged to urban areas with easy access to health facilities. Also there are differences in their life style, eating habits and social-demographic profile. Moreover this study gives a clue of nutritional status of females according to new guidelines of PNSS (Pediatrics and Pregnancy Nutrition Surveillance System). To authors knowledge there are no such studies conducted in Pakistan following these parameters to actually determine BMI status.

There is global trend towards overweight and obesity with advancing maternal age >25 years [11–13]. Our findings support this trend. This alarming accelerated trend towards overweight and obesity is multifactorial including genetic, metabolic, advancing age and to add on sedentary life style and lack of healthy diet concept with use of lot of sugar, fats, processed and junk food [11, 14, 15].

Calik, Yazdani and Neumann have shown in their studies that 78.5%, 58.5% and 39.8% overweight and obese mothers were delivered by LSCS [10, 16, 17]. Similar trend toward caesarean delivery have been shown in our study. Large size baby and abnormal fat distribution have been well documented risk factor of caesarean delivery in over-weight and obese mothers [18, 19].

**Table 3. Mean neonatal birth parameters according to maternal BMI groups.**

| Neonatal parameters | Maternal BMI (kg/m²) (n = 2766) | | | |
|---|---|---|---|---|
| | <18.5 | 18.5–24.9 | 25–29.9 | ≥ 30 |
| | (n = 110) | (1290) | (910) | (456) |
| Weight (kg) | 2.99 ± 0.68 | 3.13 ± 0.62 | 3.21 ± 0.57 | 3.24 ± 0.57 |
| Length (cm) | 46.76 ± 2.93 | 47.92 ± 2.38 | 48.18 ± 2.38 | 48.53 ± 2.46 |
| OFC (cm) | 32.86 ± 1.2) | 33.40 ± 1.2 | 33.37 ± 1.20 | 33.42 ± 1.30 |

**Table 4. Descriptive statistics of neonatal birth weight groups according to maternal BMI.**

| Neonatal birth weight | Maternal BMI (kg/m$^2$) n = 2766 (%age) | | | | |
|---|---|---|---|---|---|
| | <18.5 | 18.5–24.9 | 25–29.9 | ≥ 30 | Total |
| <2.5 | 23 (20.9%) | 184 (14.3%) | 63 (6.9%) | 27 (5.9%) | 297 (10.7%) |
| 2.5–4 | 86 (78.2%) | 1034 (80.2%) | 795 (87.4%) | 401 (87.9%) | 2316 (83.7%) |
| >4 | 1 (0.9%) | 72 (5.6%) | 52 (5.7%) | 28 (6.1%) | 153 (5.5%) |

The Persian, Russian and Korean perinatal statistics have shown more prevalence of diabetes mellitus and hypertension in overweight and obese females [16, 19, 20]. With increasing BMI above normal, there is an increase in total body fat content that leads towards metabolic derangements leading to diabetes mellitus and hypertensive disorders in pregnancy [19].

Anemia and periodontal health issues are well reported perinatal health problems in both extremes of BMI [20, 21]. Mirror image results have been reflected in our study. This is due to lack of maternal knowledge of oral hygiene, healthy diet, food taboos and food selections. Moreover, there is usual trend of poor intake of iron, folic acid, calcium and vitamin D that worsens pre-existing anemia and periodontal health issues [22].

Maternal nutritional status has effect on foetal growth in terms of weight, occipito- frontal circumference (OFC) and length. Simko et al in their study have found that risk of large size baby increases in over weight and obese mothers as OR 11.7 (95%CI 1.2–2.3) and OR 1.8 (95%CI 1.2–2.7) respectively as compared to malnourished mothers OR 0.7 (95%CI 0.5–0.7) [21]. According to Moussa et al rate of large size babies increases linearly with increasing maternal BMI [23]. According to Gondwe et al, pre-pregnancy maternal underweight increased the risk of low birth weight babies; while overweight or obesity increased the risk of large size babies [24].

Our study shows that neonatal birth growth parameters are in direct relationship with prepregnancy maternal BMI. We documented significant differences in neonatal growth parameters among 4 BMI groups. In our study, mothers with BMI <18.5 had an inclination towards low birth weight neonates and their mean weight, OFC and length were lower as compared to other BMI groups. Macrosomic neonates were seen in mothers who were overweight and obesity in their prepregnancy period.

These mirror image results are due to fact that maternal and fetal wellbeing is directly coupled. The optimal maternal nutritional status ensures continuous nutrients supply to developing foetus for growth. Maternal wellbeing and optimal nutritional status through an intricate mechanism control fetal metabolic signaling pathways and guide "fetal programming" [5–7]. Both maternal over weight and obesity result in large size neonates due to alteration in foetal metabolic programming affecting hypothalamic-pituitary axis, pancreatic islet cells and adipose tissue [10, 25]. However malnourished mothers are at high risk of delivering a small sized baby and this is because of chronic nutritional deprivation status affecting foetal growth [19].

## Conclusion

Studies determining the relationship of maternal BMI and neonatal birth weight are lacking in Pakistan. Our study highlights that there is a noteworthy direct relationship between maternal BMI and neonatal birth weight as low prepregnancy maternal BMI results in low birth weight neonates. However there is need of such more studies to highlight impact of such kind.

## Study limitations

Our study had several limitations. First of all this study lacks external validity as vast majority of mothers were from urban area. Second, paternal demographic data as not entertained in

this study which may influence neonatal growth parameters. Third, regarding maternal demographic data, we have to rely on information provided by mother as there is no national health registry system.

## Supporting information

**S1 File.**
(SAV)

## Author Contributions

**Conceptualization:** Rafia Gul.

**Data curation:** Samar Iqbal, Syed Hamza Ali, Saima Pirzada.

**Formal analysis:** Rafia Gul.

**Investigation:** Samar Iqbal, Syed Hamza Ali, Saima Pirzada.

**Methodology:** Rafia Gul.

**Project administration:** Samar Iqbal.

**Software:** Rafia Gul.

**Supervision:** Rafia Gul.

**Validation:** Zahid Anwar, Saher Gul Ahdi.

**Writing – original draft:** Rafia Gul.

**Writing – review & editing:** Zahid Anwar, Saher Gul Ahdi.

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
