## [Decision Letter · Decision Letter 0]

23 Jul 2020

PONE-D-20-17353

Pre-Pregnancy Maternal BMI As Predictor Of Neonatal Birth Weight

PLOS ONE

Dear Dr. Gul,

Thank you for submitting your manuscript to PLOS ONE. After careful consideration, we feel that it has merit but does not fully meet PLOS ONE’s publication criteria as it currently stands. Therefore, we invite you to submit a revised version of the manuscript that addresses the points raised during the review process.

This manuscript has divided opinion amongst the reviewers. I believe that the study is not without interest but it is currently in need of a major revision before it could be considered for publication.

Even before considering the reviewers' comments, It is obvious that the manuscript requires major improvement in the quality of English language used. I would therefore suggest that the authors enlist the assistance of a native English speaker to help with this. I think that of the various reviewers' comments the most important to consider is that much of the study does not seem to relate directly to its stated aim. The paper needs to be written in such a way that it develops from its aim.  In addition, it is key to present the direct association between maternal BMI (as a continuous variable) and neonatal weight (also as a continuous variable) in any revision. Finally claims of "significance" need to be removed where p>=0.05 or the 95% confidence intervals cross either side of 1.00.

All the other points raised by the reviewers need to be dealt with robustly in any revision, but the above points are of particular importance when considering whether or not any revisions can accepted for publication.

We look forward to receiving your revised manuscript.

Kind regards,

Clive J Petry, PhD

Academic Editor

PLOS ONE

Journal Requirements:

Reviewers' comments:

Reviewer's Responses to Questions

**Comments to the Author**

1. Is the manuscript technically sound, and do the data support the conclusions?

Reviewer #1: Partly

Reviewer #2: Partly

2. Has the statistical analysis been performed appropriately and rigorously? 

Reviewer #1: Yes

Reviewer #2: No

3. Have the authors made all data underlying the findings in their manuscript fully available?

Reviewer #1: Yes

Reviewer #2: Yes

4. Is the manuscript presented in an intelligible fashion and written in standard English?

Reviewer #1: Yes

Reviewer #2: Yes

5. Review Comments to the Author

Reviewer #1: I want to congratulate the authors for the effort in writing this interesting paper. Overall, I think the paper needs to improve the grammar and the fluency of the reading specially in the introduction section. I have other minor comments as well.

Introduction

- I'm not sure if maternal BMI measures body fat; instead, it is a ratio between weight and height, can you please include the reference of this sentence?

- It would be interesting to know the consequences of pre-pregnancy BMI on the offspring health

- Authors mention "fetal programming" but I would be interesting to know the conection between programming and pre-pregnancy BMI, at least in general.

- I'm not sure if the last sentence of the introduction section is related with the aim of the study, e.g. the study evaluates preventive measures?

Materials and methods

- Why authors did not consider materna weight gain during pregnancy as a newborn weight factor?

- I consider it is important to know the method to take the maternal measures as weight and height as well as the method to take the newborn weight.

- Why the authors did not include newborn height, not even as a characterisation variable?

- Medical records are electronic, hand writing, etc?

- How did the authors verifies the quality of data?

Results

- I don't understand the associations in table 1, I consider it need some legend at the end to clarify the comparisons between significant associations.

- City and village mean urbal and rural?

- As the main outcome is the birth weight, tables do not reflect that association. It seems like authors made associations with maternal BMI as the main outcome.

- I strongly recommend the authors to show the association between maternal BMI and birth weight, as it is the aim of the study.

Discussion

- It should be mainly about the association between birth weight and maternal BMI and not about the factors that affect maternal BMI

Reviewer #2: The results are not consistent with the purpose of the study. (Relation between pre-pregnancy BMI and neonatal birth weight, whereas majority of results concern on anemia, hypertension and diabetes mellitus).

Result are not statistically significant in many cases although authors state that they are.

In methods study is descripted as prospective, descriptive. In discussion study is descripted as retrospective, cross-sectional. It should be unified.

There is lack of information about excluded cases.

Table 1 show only that there is a lot of confounding factors, there is no possibility to form conclusion on it. There is lack of post-hoc tests. Authors made 2 categorizes of overweight and obese but then reported in as a one group >25kg/m2.

Table 2 is unclear. It related to risk factor of what in pregnancy? I can guess that it is risk factor of hypotrophy or macrosomia but I'm not sure. It should be explained what mean medical illnesses, uterine problems. I don't know why singleton pregnancy is a risk factor. According to WHO appropriate gestational weight gain is different for underweight, normal weight, overweight and obese. So 10-15 kg is not correct value.

The most important calculation which concern BMI and birth weight should be in table, no in graph without any statistics, not only in text. Also raw date about newborns length and occipital frontal circumferences give us no information.

In introduction there is a lot about malnutrition, but then authors conclude that in Pakistan there is much higher problem with overweight and obese, may be introduction should be rewritten.

Discussion is unclear, hard to read, not interesting.

There is lack of unit in BMI

There is lack of abbreviation explanation (SVD, LSCS, OFC)

6. PLOS authors have the option to publish the peer review history of their article (what does this mean?). If published, this will include your full peer review and any attached files.

Reviewer #1: **Yes: **Fanny Aldana-Parra

Reviewer #2: No

---

## [Author Response · Author response to Decision Letter 0]

16 Sep 2020

REBUTTAL LETTER

Date: Sep 06, 2020 21.00 pm

To: “PLOS ONE" plosone@plos.org"

From: Rafia Gul" docrafiagul@gmail.com

Subject: Response to reviewers

PONE-D-20-17353

Title: Pre-Pregnancy Maternal BMI as Predictor of Neonatal Birth Weight

Respected Petry and Reviewers, 

It’s really an honor for me that your esteemed journal considered my article for review and highlighted things that required revision. Keeping in consideration your reviews, many changes have been made in article and have been highlighted as per your requirement.

COMMENTS: 

Even before considering the reviewers' comments, It is obvious that the manuscript requires major improvement in the quality of English language used. I would therefore suggest that the authors enlist the assistance of a native English speaker to help with this. I think that of the various reviewers' comments the most important to consider is that much of the study does not seem to relate directly to its stated aim. The paper needs to be written in such a way that it develops from its aim. In addition, it is key to present the direct association between maternal BMI (as a continuous variable) and neonatal weight (also as a continuous variable) in any revision. Finally claims of "significance" need to be removed where p>=0.05 or the 95% confidence intervals cross either side of 1.00.

ANSWER:

• Quality of English has been improved with help of native English speaker to make it simpler and more understandable.

• Impact of maternal BMI on neonatal weight has been presented in more detail using statistical analysis

• Corrections in interpretation of statistically significant results have been done.

COMMENTS:

ANSWERS:

• A rebuttal letter has been attached under file name “ Response to Reviewers”

• A marked up copy that highlights changes made to original version labeled as 'Revised Manuscript with Track Changes'. has been uploaded

• An unmarked version of our revised paper without tracked changes labeled as “ Manuscript” has been uploaded 

COMMENTS:

ANSWER:

No changes to your financial disclosure

COMMENTS:

ANSWERS:

Not applicable

Journal Requirements:

COMMENTS:

1. Please ensure that your manuscript meets PLOS ONE's style requirements, including those for file naming. The PLOS ONE style templates can be found at, https://journals.plos.org/plosone/s/file?id=wjVg/PLOSOne_formatting_sample_main_body.pdf and https://journals.plos.org/plosone/s/file?id=ba62/PLOSOne_formatting_sample_title_authors_affiliations.pdf

ANSWER:

Manuscript has been organized according to PLOS ONE's style requirements

ANSWER: 

Data for review has been made available 

REVIEWERS' COMMENTS:

Reviewer's Responses to Questions

Reviewer #1: 

COMMENTS:

I want to congratulate the authors for the effort in writing this interesting paper. Overall, I think the paper needs to improve the grammar and the fluency of the reading specially in the introduction section. I have other minor comments as well.

ANSWER:

Thank you so much for your kind response. I have tried to improve grammar and fluency to make it more understandable.

COMMENTS: Introduction

- I'm not sure if maternal BMI measures body fat; instead, it is a ratio between weight and height, can you please include the reference of this sentence?

- It would be interesting to know the consequences of pre-pregnancy BMI on the offspring health

- Authors mention "fetal programming" but I would be interesting to know the conection between programming and pre-pregnancy BMI, at least in general.

- I'm not sure if the last sentence of the introduction section is related with the aim of the study, e.g. the study evaluates preventive measures?

ANSWERS:

• BMI has been redefined in introduction

• Introductory part has been revised with addition of literature highlighting the consequences of pre-pregnancy BMI on the offspring health

• Concept of "fetal programming" has been eplined in more detail

• Aim of study has been rephrased. 

COMMENTS: Materials and methods

- Why authors did not consider materna weight gain during pregnancy as a newborn weight factor?

- I consider it is important to know the method to take the maternal measures as weight and height as well as the method to take the newborn weight.

- Why the authors did not include newborn height, not even as a characterisation variable?

- Medical records are electronic, hand writing, etc?

- How did the authors verifies the quality of data?

ANSWERS:

• Maternal weight gain during pregnancy was part of original study. It had been mentioned as confounder factor of study.

• Methodology has been described in more detail regarding data collection format, measuring gadgets and persons involved. 

• All neonatal birth parameters weight, length and OFC have been described in more detail.

COMMENTS : Results

- I don't understand the associations in table 1, I consider it need some legend at the end to clarify the comparisons between significant associations.

- City and village mean urbal and rural?

- As the main outcome is the birth weight, tables do not reflect that association. It seems like authors made associations with maternal BMI as the main outcome.

- I strongly recommend the authors to show the association between maternal BMI and birth weight, as it is the aim of the study.

ANSWER:

• Necessary changes have been made in table 1 to make more understandable

• Yes, city = urban , village = rural

• Impact of maternal BMI and neonatal birth weight has been statistically highlighted using ANOVA tests, post hoc analysis and multivariate regression analysis.

COMMENTS: Discussion

- It should be mainly about the association between birth weight and maternal BMI and not about the factors that affect maternal BMI

ANSWER:

Necessary changes have been made in discussion.

Reviewer #2: 

COMMENTS:

1. The results are not consistent with the purpose of the study. (Relation between pre-pregnancy BMI and neonatal birth weight, whereas majority of results concern on anemia, hypertension and diabetes mellitus). Result are not statistically significant in many cases although authors state that they are.

2. In methods study is descripted as prospective, descriptive. In discussion study is descripted as retrospective, cross-sectional. It should be unified.

3. There is lack of information about excluded cases.

4. Table 1 show only that there is a lot of confounding factors, there is no possibility to form conclusion on it. 

5. There is lack of post-hoc tests. 

6. Authors made 2 categorizes of overweight and obese but then reported in as a one group >25kg/m2.

7. Table 2 is unclear. It related to risk factor of what in pregnancy? I can guess that it is risk factor of hypotrophy or macrosomia but I'm not sure. It should be explained what mean medical illnesses, uterine problems. I don't know why singleton pregnancy is a risk factor. 

8. According to WHO appropriate gestational weight gain is different for underweight, normal weight, overweight and obese. So 10-15 kg is not correct value.

9. The most important calculation which concern BMI and birth weight should be in table, no in graph without any statistics, not only in text. Also raw date about newborns length and occipital frontal circumferences give us no information.

10. In introduction there is a lot about malnutrition, but then authors conclude that in Pakistan there is much higher problem with overweight and obese, may be introduction should be rewritten.

11. Discussion is unclear, hard to read, not interesting.

12. There is lack of unit in BMI

13. There is lack of abbreviation explanation (SVD, LSCS, OFC)

ANSWERS:

1. Results have been rationalized

2. Study design has been unified

3. Excluded cases have been described in more detail

4. Table 1 has been tabulated in better way

5. Post hoc test have been added

6. Overweight and obese mothers have been categorized and results for each group have been described for each group separately

7. Table 2 has been omitted for purpose of clearity

8. According to WHO appropriate gestational weight gain for each BMI group has been highlighted and results have been calculated accordingly.

9. Association of BMI and birth weight has been tabulated.

10. Introductory part has been rewritten with focus on overweight and obesity

11. BMI units as kg/m2 has been mentioned as per requirement

12. Abbreviations have been explained

Regards

Rafia Gul

---

## [Decision Letter · Decision Letter 1]

2 Oct 2020

Pre-Pregnancy Maternal BMI As Predictor Of Neonatal Birth Weight

PONE-D-20-17353R1

Dear Dr. Gul,

We’re pleased to inform you that your manuscript has been judged scientifically suitable for publication and will be formally accepted for publication once it meets all outstanding technical requirements.

Kind regards,

Clive J Petry, PhD

Academic Editor

PLOS ONE

Additional Editor Comments (optional):

Reviewers' comments:

Reviewer's Responses to Questions

**Comments to the Author**

1. If the authors have adequately addressed your comments raised in a previous round of review and you feel that this manuscript is now acceptable for publication, you may indicate that here to bypass the “Comments to the Author” section, enter your conflict of interest statement in the “Confidential to Editor” section, and submit your "Accept" recommendation.

Reviewer #1: All comments have been addressed

2. Is the manuscript technically sound, and do the data support the conclusions?

Reviewer #1: Yes

3. Has the statistical analysis been performed appropriately and rigorously? 

Reviewer #1: Yes

4. Have the authors made all data underlying the findings in their manuscript fully available?

Reviewer #1: Yes

5. Is the manuscript presented in an intelligible fashion and written in standard English?

Reviewer #1: Yes

6. Review Comments to the Author

Reviewer #1: I consider the authors addressed the majority of my comments and the comments made by other reviewers, I consider that the paper can be published.

7. PLOS authors have the option to publish the peer review history of their article (what does this mean?). If published, this will include your full peer review and any attached files.

Reviewer #1: **Yes: **Fanny Aldana-Parra

---

## [Editor Report · Acceptance letter]

8 Oct 2020

PONE-D-20-17353R1 

Pre-pregnancy maternal BMI as predictor of neonatal birth weight 

Dear Dr. Gul:

I'm pleased to inform you that your manuscript has been deemed suitable for publication in PLOS ONE. Congratulations! Your manuscript is now with our production department. 

Kind regards, 

on behalf of

Dr. Clive J Petry 

Academic Editor

PLOS ONE